# A Cohort of Patients with COVID-19 in a Major Teaching Hospital in Europe

**DOI:** 10.3390/jcm9061733

**Published:** 2020-06-04

**Authors:** Alberto M. Borobia, Antonio J. Carcas, Francisco Arnalich, Rodolfo Álvarez-Sala, Jaime Monserrat-Villatoro, Manuel Quintana, Juan Carlos Figueira, Rosario M. Torres Santos-Olmo, Julio García-Rodríguez, Alberto Martín-Vega, Antonio Buño, Elena Ramírez, Gonzalo Martínez-Alés, Nicolás García-Arenzana, M. Concepción Núñez, Milagros Martí-de-Gracia, Francisco Moreno Ramos, Francisco Reinoso-Barbero, Alejandro Martin-Quiros, Angélica Rivera Núñez, Jesús Mingorance, Carlos J. Carpio Segura, Daniel Prieto Arribas, Esther Rey Cuevas, Concepción Prados Sánchez, Juan J. Rios, Miguel A. Hernán, Jesús Frías, José R. Arribas

**Affiliations:** 1Clinical Pharmacology Department, La Paz University Hospital-IdiPAZ, Universidad Autónoma de Madrid, 28046 Madrid, Spain; jaime.monserrat@salud.madrid.org (J.M.-V.); elena.ramirezg@uam.es (E.R.); jesus.frias@uam.es (J.F.); 2Internal Medicine Department, La Paz University Hospital-IdiPAZ, Universidad Autónoma de Madrid, 28046 Madrid, Spain; farnalich@salud.madrid.org (F.A.); juanj.rios@salud.madrid.org (J.J.R.); 3Pneumology Department, La Paz University Hospital-IdiPAZ, Universidad Autónoma de Madrid, 28046 Madrid, Spain; rodolfo.alvarezsala@salud.madrid.org (R.Á.-S.); carlosjavier.carpio@salud.madrid.org (C.J.C.S.); m.concepcion.prados@salud.madrid.org (C.P.S.); 4Intensive Care Unit, La Paz University Hospital-IdiPAZ, Universidad Autónoma de Madrid, 28046 Madrid, Spain; manuel.quintana@salud.madrid.org (M.Q.); juancarlos.figueira@salud.madrid.org (J.C.F.); 5Emergency Department, La Paz University Hospital-IdiPAZ, Universidad Autónoma de Madrid, 28046 Madrid, Spain; rosario.torres@salud.madrid.org (R.M.T.S.-O.); amquiros@salud.madrid.org (A.M.-Q.); angelica.rivera@salud.madrid.org (A.R.N.); 6Microbiology Department, La Paz University Hospital-IdiPAZ, 28046 Madrid, Spain; jgarciarodriguez@salud.madrid.org (J.G.-R.); jesus.mingorance@idipaz.es (J.M.); 7CSUR Coordination, La Paz University Hospital-IdiPAZ, 28046 Madrid, Spain; amartinv@salud.madrid.org; 8Laboratory Medicine Department, La Paz University Hospital-IdiPAZ, 28046 Madrid, Spain; antonio.buno@salud.madrid.org (A.B.); dprietoa@salud.madrid.org (D.P.A.); 9Department of Epidemiology, Columbia University Mailman School of Public Health, New York, NY 10032, USA; gm2794@cumc.columbia.edu; 10Preventive Medicine Department, La Paz University Hospital-IdiPAZ, 28046 Madrid, Spain; ngarciaarenzana@salud.madrid.org; 11Risk Prevention Department, La Paz University Hospital-IdiPAZ, 28046 Madrid, Spain; mnunezl@salud.madrid.org; 12Emergency Radiology Unit, La Paz University Hospital-IdiPAZ, 28046 Madrid, Spain; milagros.marti@salud.madrid.org; 13Pharmacy Department, La Paz University Hospital-IdiPAZ, 28046 Madrid, Spain; fmorenor.hulp@salud.madrid.org; 14Anesthesiology Department, La Paz University Hospital-IdiPAZ, 28046 Madrid, Spain; francisco.reinoso@salud.madrid.org; 15Nursing Department, La Paz University Hospital-IdiPAZ, 28046 Madrid, Spain; esther.rey@salud.madrid.org; 16Departments of Epidemiology and Biostatistics, Harvard T.H. Chan School of Public Health, Harvard-MIT Division of Health Sciences and Technology, Boston, MA 02115, USA; mhernan@hsph.harvard.edu

**Keywords:** COVID19, SARS-CoV-2, Spain, Europe

## Abstract

Background: Since the confirmation of the first patient infected with SARS-CoV-2 in Spain in January 2020, the epidemic has grown rapidly, with the greatest impact on the region of Madrid. This article describes the first 2226 adult patients with COVID-19, consecutively admitted to La Paz University Hospital in Madrid. Methods: Our cohort included all patients consecutively hospitalized who had a final outcome (death or discharge) in a 1286-bed hospital of Madrid (Spain) from 25 February (first case admitted) to 19 April 2020. The data were manually entered into an electronic case report form, which was monitored prior to the analysis. Results: We consecutively included 2226 adult patients admitted to the hospital who either died (460) or were discharged (1766). The patients’ median age was 61 years, and 51.8% were women. The most common comorbidity was arterial hypertension (41.3%), and the most common symptom on admission was fever (71.2%). The median time from disease onset to hospital admission was 6 days. The overall mortality was 20.7% and was higher in men (26.6% vs. 15.1%). Seventy-five patients with a final outcome were transferred to the intensive care unit (ICU) (3.4%). Most patients admitted to the ICU were men, and the median age was 64 years. Baseline laboratory values on admission were consistent with an impaired immune-inflammatory profile. Conclusions: We provide a description of the first large cohort of hospitalized patients with COVID-19 in Europe. Advanced age, male sex, the presence of comorbidities and abnormal laboratory values were more common among the patients with fatal outcomes.

## 1. Introduction

As of this writing, Spain has the second highest number of confirmed severe acute respiratory coronavirus 2 (SARS-CoV-2) infections worldwide after the United States. The first infection in Spain was confirmed on 31 January 2020, in La Gomera in the Canary Islands [1]. In the Madrid region, the first infection was registered on 25 February 2020. Madrid is a densely populated area with 6.7 million inhabitants [2] and has felt the greatest impact from the pandemic in Spain. The number of confirmed cases in Madrid was 58,819 as of 25 April (26.3% of cases in Spain) [3], reaching a peak of 3419 new cases on 30 March.

The progression of the outbreak in Madrid is similar to that observed in the most affected areas in Western countries, such as the Lombardy region in Italy and New York City in the US. The healthcare systems of these regions are under massive stress, and the cumulative COVID-19 mortality per 100,000 inhabitants since the start of the pandemic has been high: 132 deaths in Lombardy, 140 in New York City and 190 in Madrid (as of 25 April) [4,5,6].

The La Paz University Hospital is a large teaching hospital with a catchment area of 527,366 inhabitants in the north of Madrid. Shortly after the outbreak, the hospital’s procedures were adapted to cope with the rise in COVID-19 cases. By 25 April, the hospital had admitted over 2500 patients with COVID-19, one of the largest single-site cohorts in Europe. These patients’ clinical information was collected using a standardized protocol. 

In this article, we describe the first 2226 adult patients consecutively admitted with a confirmed diagnosis of SARS-CoV-2 infection to La Paz University Hospital and who had died or been discharged by 19 April.

## 2. Methods

### 2.1. Study Population

Our study included all individuals, 18 years or older, who were hospitalized in the wards (or emergency department, due to the lack of available beds in the wards) of La Paz University Hospital with a diagnosis of COVID-19 and who either died or were discharged by 19 April. Patients discharged from the emergency department after a stay of less than 24 h were not considered hospitalized and were not included in this analysis.

### 2.2. Data Collection

We employed a modified version of the electronic case record form (eCRF) for severe acute respiratory infections, developed by the World Health Organization/International Severe Acute Respiratory and Emerging Infection Consortium [7]. Our eCRF includes 372 variables, grouped into demographics, medical history, infection-exposure history, symptoms, complications, treatments (excluding clinical trials) and disease progression during hospitalization (see Appendix A).

We collected the clinical data directly extracting the information from the hospital’s database, when possible, or by a manual and individual review of the patients’ electronic clinical records, including the clinical notes (DXC-HCIS- Healthcare Information System). The clinical data collected at hospital admission included age, sex, smoking status, transmission, comorbidities, symptoms on admission, respiratory status and time from disease onset. Complications during hospitalization and intensive care unit (ICU) admission were also recorded. The data collection effort was conducted by a volunteer team of resident doctors and senior medical students. Data monitoring was conducted by our hospital’s Central Clinical Research Unit.

Laboratory results (hematology, biochemistry, microbiology) were extracted from various hospital data management systems, and information regarding the drugs used during hospitalization was extracted from the electronic prescription system. 

### 2.3. Statistical Analysis

Continuous variables are presented as means and standard deviations (SD) or medians and interquartile ranges (IQR), and categorical variables are listed as numbers and percentages (%). To analyze predictors of in-hospital death, we employed a multivariate logistic regression model. We selected the variables for inclusion in the model on the basis of previous findings and considering the total number of deaths in our study to avoid overfitting the model. In a sensitivity analysis to explore potential collider bias, we restricted the fit of the model to patients admitted before 20 March(results did not materially change). We performed the statistical calculations using R (version 3.4.0) [8].

The study was approved by the Research Ethics Committee of La Paz University Hospital (PI-4072) and by the Spanish Agency of Medicines and Medical Devices (HUL-AIN-2020-01) and was registered in the European Union Electronic Register of Post-Authorization Studies (EUPAS34331).

## 3. Results

A total of 3127 patients were consecutively treated in the emergency department of La Paz University Hospital between 25 February and 19 April 2020. Of these, 2226 adult patients were hospitalized and either died (460, 20.7%) or were discharged (1766, 79.3%) and were therefore included in our analysis. Figure 1 shows the bed occupancy by patients with COVID-19 over time, with a peak of 1033 beds, 106 of which were in the ICU (compared with 30 ICU beds before the COVID-19 pandemic).

The median time from clinical onset to hospital admission was 6 days (IQR, 3–9). At admission, the patients had a median age of 61 (IQR, 46–78) years, 52% were women, 41% had arterial hypertension, 19% had chronic heart disease, and 17% had diabetes mellitus. The most common symptoms at admission were fever, cough and dyspnea, and the median oxygen saturation at admission was 95% (IQR, 92–97). The most common complications during hospitalization were acute confusional syndrome, acute kidney failure and acute respiratory distress syndrome (Table 1).

Most of the patients were treated with drugs presumed to have antiviral activity against SARS-CoV-2. The most frequent combination was hydroxychloroquine plus azithromycin followed by hydroxychloroquine in isolation.

At the time of the analysis, 237 patients had been admitted to the ICU, 116 remained in the ICU, 55 had died, 20 were discharged from the hospital, and 46 remained confined to a standard hospital bed. Table 2 shows the demographic characteristics, comorbidities and respiratory status on the day of emergency department admission of the 75 patients (3.4%) transferred to the ICU who had died or been discharged by 19 April. Compared with the entire cohort, the patients admitted to the ICU were older (median age, 64 vs. 61 years), had a higher male/female ratio (3.2 vs. 0.93) and had a higher prevalence of arterial hypertension (52 vs. 41.3%), obesity (30.7% vs. 10.9%), diabetes mellitus (28.0% vs. 17.1%) and chronic obstructive pulmonary disease (17.3% vs. 6.9%).

Table 3 shows the mortality by age group and sex for the 2226 patients. The overall mortality was 26.6% for the men and 15.1% for the women. Mortality increased with age, reaching over 60% for patients over 80 years of age.

Table 4 shows the laboratory findings at admission for the entire cohort and for the ICU subgroup. In the cohort, the baseline creatine kinase, creatinine, D-dimer, ferritin, lactate dehydrogenase, procalcitonin, C-reactive protein and high-sensitivity cardiac troponin I levels and prothrombin times were higher among the non-survivors than the survivors. In the cohort admitted to the ICU, the most notable differences with the whole cohort were higher levels of D-dimer, ferritin, C reactive protein and troponin. Within the ICU cohort, the lymphocyte counts and procalcitonin and C-reactive protein levels at hospital admission were also clearly higher in the patients who died compared with those who survived.

In the multivariable logistic regression model, we found that male sex, older age, an oxygen saturation <90% on admission, lower lymphocyte count and high C-reactive protein levels were associated with a high probability of death (Table 5).

## 4. Discussion

To the best of our knowledge, this is the first report of a large cohort of patients hospitalized with COVID-19 in Europe. Our cohort includes all patients with a final outcome (discharge or death) consecutively admitted to our hospital during the worst phase of COVID-19 in Madrid’s hospital system.

Similar to other cohorts [9,10,11,12,13,14,15], the hospitalized patients in Madrid were elderly and had numerous comorbidities, the most common of which were arterial hypertension and diabetes. Our male/female ratio was 0.9, which is lower than the 1.5 reported in the series from Wuhan (China) [12] and New York City [10]. In Madrid, the male/female ratio for individuals older than 60 and 75 years was 0.74 and 0.61, respectively [16]. Differences compared with other cohorts could be partially explained by the different male/female ratio in the Madrid population pyramid. Despite our lower male/female ratio compared with other reports, the mortality for each age group was notably higher for the male patients than for the female patients, as reported in other cohorts. It is relevant that one third of the patients included in our cohort were nursing homes residents.

The overall mortality (20.7%) in our series by age group was similar to that of the New York cohort (21%) [10] and lower than that of a Wuhan cohort (28.3%) [12]. In our cohort, older age and the presence of comorbidities were more common among the patients with fatal outcomes, both for the entire cohort and for those admitted to the ICU.

The most frequent symptoms at admission were fever, cough and dyspnea; myalgia and diarrhea were also common. Notably, 12.8% of the patients had anosmia as a presenting symptom, as described in other cohorts [17]. The time from disease onset was short (6 days), and the patients’ respiratory status on admission (as reflected by oxygen saturation) was generally poor, with half of the patients presenting an oxygen saturation of < 95%. General symptoms, such as diarrhea, myalgia, headache and anosmia, were more common among survivors, while respiratory symptoms such as dyspnea and sputum production were more prevalent among the non-survivors, who also had a lower median oxygen saturation on admission (90% vs. 96%). In any case, these findings need to be confirmed in other large cohorts and meta-analyses.

The laboratory values on admission were consistent with an impaired immune-inflammatory profile, characterized by lymphopenia and elevated D-dimer, procalcitonin, ferritin and C-reactive protein levels. Most of these abnormal laboratory readings were more common in the patients with fatal outcomes. Creatine kinase and troponin levels on admission were also higher in the patients with fatal outcomes, a finding also reported in other series [18].

Most of our patients underwent treatments presumed to have activity against SARS-CoV-2. The indication for treatment in Spain has changed during the course of the pandemic but has always been based on the indications by the Spanish Ministry of Health and the availability of treatments. Our hospital has also participated in several clinical trials, the results of which we cannot provide in this manuscript (remdesivir, tocilizumab, sarilumab). We have included in this article the use of treatments with potentially antiviral effects available in our hospital but that have not been included in a clinical trial (hydroxychloroquine, azithromycin and lopinavir/ritonavir). It is also noteworthy that, at the start of the pandemic in Spain, only symptomatic treatment was indicated. At present, there are no clinical trial data that support the use of any of these treatments for improving the outcomes of patients with COVID-19.

This study has a number of limitations. First, the data were collected from various databases, both manually and automatically. The data manually entered into the eCRF were monitored and curated. Second, we did not conduct a follow-up of the patients after discharge. Third, our reported mortality rates might change once the entire cohort of hospitalized patients has been analyzed.

In summary, this study provides initial data on the clinical and laboratory features and outcomes of patients hospitalized with COVID-19 infection in a large teaching hospital in Madrid during the peak of the pandemic in Spain.

## Figures and Tables

**Figure 1 jcm-09-01733-f001:**
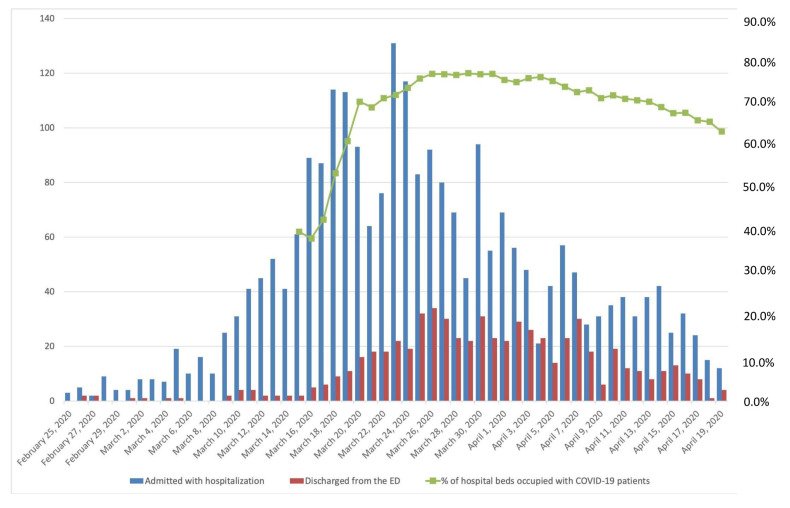
Patients treated by the emergency department per day between 25 February 2020, and 19 April 2020. The blue lines indicate patients admitted and hospitalized, and the red lines indicate patients discharged from the emergency department. The green line represents the percentage of hospital beds (including ICU) occupied by patients with COVID-19.

**Table 1 jcm-09-01733-t001:** Characteristics at admission and complications during hospitalization.

Variable	Total	Deaths	Live Discharges	*p*-Value
	(*N* = 2226)	(*N* = 460)	(*N* = 1766)	
Median age, years (IQR)	61 (46–78)	82.5 (76–87)	56 (42–71)	< 0.001 (w)
Sex, *n* (%)				< 0.001
Male	1074 (48.2)	286 (62.2)	788 (44.6)	
Female	1152 (51.8)	174 (37.8)	978 (55.4)	
Nursing home resident, *n* (%)	709 (31.9)	125 (27.2)	584 (33.1)	0.018
Current smokers, *n* (%)	157 (7.1)	44 (9.6)	113 (6.4)	0.023
Comorbidities (at least one), *n* (%)	1747 (78.5)	448 (97.4)	1299 (73.6)	< 0.001
Arterial hypertension	920 (41.3)	318 (69.1)	602 (34.1)	< 0.001
Chronic heart disease	429 (19.3)	195 (42.4)	234 (13.3)	< 0.001
Diabetes mellitus	381 (17.1)	157 (34.1)	224 (12.7)	< 0.001
Rheumatological disease	268 (12.0)	80 (17.4)	188 (10.6)	< 0.001
Solid malignant disease	252 (11.3)	93 (20.2)	159 (9.0)	< 0.001
Obesity	242 (10.9)	66 (14.3)	176 (10.0)	0.009
Chronic kidney disease	174 (7.8)	94 (20.4)	80 (4.5)	< 0.001
Chronic obstructive pulmonary disease	153 (6.9)	65 (14.1)	88 (5.0)	< 0.001
Other chronic lung diseases	143 (6.4)	49 (10.7)	94 (5.3)	< 0.001
Hematological malignant disease	133 (6.0)	46 (10.0)	87 (4.9)	< 0.001
Asthma	115 (5.2)	17 (3.7)	98 (5.5)	0.138
Liver disease	89 (4.0)	23 (5.0)	66 (3.7)	0.272
HIV infection	13 (0.6)	4 (0.9)	9 (0.5)	0.3218 (f)
Symptoms on admission, *n* (%)				
Fever	1585 (71.2)	325 (70.7)	1260 (71.3)	0.814
Cough	1373 (61.7)	228 (49.6)	1145 (64.8)	< 0.001
Dyspnea	1108 (49.8)	278 (60.4)	830 (47.0)	< 0.001
Myalgia	596 (26.8)	51 (11.1)	545 (30.9)	< 0.001
Diarrhea	484 (21.7)	62 (13.5)	422 (23.9)	< 0.001
Headache	427 (19.2)	24 (5.2)	403 (22.8)	< 0.001
Sputum production	320 (14.4)	80 (17.4)	240 (13.6)	0.046
Nausea or vomiting	299 (13.4)	42 (9.1)	257 (14.6)	0.003
Anosmia	284 (12.8)	5 (1.1)	279 (15.8)	0.003
Respiratory status				
Median SatO_2_ on admission, % (IQR)	95 (92–97)	90 (85–93.3)	96 (93–97)	< 0.001 (w)
SatO_2_ < 90% on admission, *n* (%)	234 (10.5)	126 (27.4)	108 (6.1)	< 0.001
Median time from disease onset to hospital admission, days (IQR)	6 (3–9)	4 (2–7)	7 (3–10)	< 0.001 (w)
Complications during hospitalization				
Acute confusional syndrome	197 (8.8)	149 (32.4)	48 (2.7)	< 0.001
Acute kidney failure	173 (7.8)	120 (26.1)	53 (3.0)	< 0.001
Acute respiratory distress syndrome	109 (4.9)	94 (20.4)	15 (0.8)	< 0.001
Bacterial pneumonia	64 (2.9)	45 (9.8)	19 (1.1)	< 0.001
Acute heart failure	51 (2.3)	34 (7.4)	17 (1.0)	< 0.001
Arrhythmia	46 (2.1)	33 (7.2)	12 (0.7)	< 0.001
Treatment regimens, *n* (%)				
Hydroxychloroquine + azithromycin	706 (31.7)	180 (39.1)	526 (29.8)	< 0.001
Hydroxychloroquine	545 (24.5)	163 (35.4)	382 (21.6)	< 0.001
Hydroxychloroquine + lopinavir/ritonavir	116 (5.2)	23 (5.0)	93 (5.3)	0.911
Hydroxychloroquine + azithromycin + lopinavir/ritonavir	69 (3.1)	23 (5.0)	46 (2.6)	0.012
Lopinavir/ritonavir	19 (0.8)	3 (0.6)	16 (0.9)	0.216 (f)

Abbreviations: (f), Fisher’s Exact Test; HIV, human immunodeficiency virus; IQR, interquartile range; SatO_2_, oxygen saturation; (w), Wilcoxon Rank Sum Test.

**Table 2 jcm-09-01733-t002:** Demographics, Comorbidities and Respiratory Status on Admission of the Intensive Care Patients.

Variable	Total	Deaths	Live Discharges	*p*-Value
	(*N* = 75)	(*N* = 55)	(*N* = 20)	
Median age, years (IQR)	64 (54–71)	69 (62–72.5)	47 (38.5–56)	< 0.001 (w)
Sex, *n* (%)				0.127 (f)
Male	57 (76)	39 (70.9)	18 (90)	
Female	18 (24)	16 (29.1)	2 (10)	
Current smokers, *n* (%)	10 (13.3)	7 (12.7)	3 (15)	1.000 (f)
Comorbidities (at least one), *n* (%)	67 (89.3)	52 (94.6)	15 (75)	0.045
Arterial hypertension	39 (52.0)	32 (58.2)	7 (35)	0.130
Obesity	23 (30.7)	18 (32.7)	5 (25)	0.719
Diabetes mellitus	21 (28.0)	17 (30.9)	4 (20)	0.401 (f)
Chronic obstructive pulmonary disease	13 (17.3)	12 (21.8)	1 (5)	0.165 (f)
Chronic heart disease	11 (14.7)	10 (18.2)	1 (5)	0.269 (f)
Solid malignant disease	9 (12.0)	6 (10.9)	3 (15)	0.693 (f)
Rheumatological disease	7 (9.3)	5 (9.1)	2 (10)	1.000 (f)
Chronic kidney disease	5 (6.7)	5 (9.1)	0 (0)	0.316 (f)
Asthma	4 (5.3)	3 (5.5)	1 (5)	1.000 (f)
Other chronic lung diseases	4 (5.3)	4 (7.3)	0 (0)	0.568 (f)
Liver disease	3 (4.0)	2 (3.6)	1 (5)	1.000 (f)
Hematological malignant disease	3 (4.0)	3 (5.5)	0 (0.0)	0.560 (f)
HIV infection	1 (1.3)	1 (1.8)	0 (0)	1.000 (f)
Respiratory status, *n* (%)				
SatO_2_ < 90% on admission	11 (17)	9 (16.3)	2 (10)	0.717 (f)

Abbreviations: (f), Fisher’s Exact Test; HIV, human immunodeficiency virus; IQR, interquartile range; SatO_2_, oxygen saturation; (w), Wilcoxon Rank Sum Test.

**Table 3 jcm-09-01733-t003:** Mortality distribution by age group and sex.

Age Group	Male		Female		Total	
	*N*	Mortality, %	*N*	Mortality, %	*N*	Mortality, %
18–29 years	59	0.0	100	1.0	159	0.6
30–39 years	86	0.0	133	0.0	219	0
40–49 years	120	1.7	167	1.2	287	1.5
50–59 years	166	4.8	199	3.0	365	3.8
60–69 years	120	20.0	152	7.9	327	11
70–79 years	202	41.1	156	25.0	358	34.1
80–89 years	212	62.7	179	41.3	391	52.9
≥90 years	54	66.7	66	60.6	120	63.3

**Table 4 jcm-09-01733-t004:** Laboratory findings on admission.

Variable, Median (IQR)	*N* with Data	Total	Deaths	Live Discharges	*p*-Value
Entire cohort		*n* = 2226	*n* = 460	*n* = 1766	
Lymphocyte count,	1805	0.93 (0.91–0.97)	0.68 (0.64–0.74)	1.02 (0.99–1.06)	< 0.001 (w)
×10⁹/L					
(NR, 1.1–4.5)					
ALT, U/L	1526	31 (30–33)	28 (27–31)	32 (31–34)	< 0.001 (w)
(NR, 0–35)					
Albumin g/dL	1200	4.2 (4.2–4.3)	3.9 (3.9–4)	4.3 (4.3–4.4)	< 0.001 (w)
(NR, 2.9–5.2)					
Creatine kinase, U/L	1241	100 (95–105)	143 (121–164)	88 (83–95)	< 0.001 (w)
(NR, 40–280)					
Creatinine mg/dL	1602	0.81 (0.8–0.83)	1.05 (0.99–1.13)	0.77 (0.75–0.78)	< 0.001 (w)
(NR, 0.7–1.3)					
D-dimer, ng/mL	982	720 (680–758)	1590 (1309–1952)	608 (560–648)	< 0.001 (w)
(NR, 0–500)					
Serum ferritin, ng/L	599	346 (312–393)	588 (490–741)	313 (261–358)	< 0.001 (w)
(NR, 22–322)					
Platelet count, ×10⁹/L	1605	218 (213–225)	198 (186–206)	226 (219–234)	< 0.001 (w)
(NR, 150–370)					
IL-6, pg/mL	90	28.4 (18.8–40.1)	30.9 (27.2–244)	26 (17.9–43.9)	< 0.251 (w)
(NR, 0–3.4)					
Lactate dehydrogenase, U/L (NR, 100–190)	1458	321 (315–330)	380 (364–408)	308 (301–316)	< 0.001 (w)
Procalcitonin, ng/mL	637	0.12 (0.1–0.14)	0.4 (0.3–0.5)	0.09 (0.08–0.11)	< 0.001 (w)
(NR, 0–0.5)					
C-reactive protein, mg/L	1482	71.1 (65.6–77.4)	137.7 (118.1–153.8)	57.8 (51.8–62)	< 0.001 (w)
(NR, 0–5)					
Prothrombin time, s	1581	11.1 (11.1–11.2)	11.5 (11.4–11.7)	10.9 (10.9–11)	< 0.001 (w)
High-sensitivity cardiac	253	11.2 (9.1–18.2)	66.9 (29.7–115.2)	8.2 (6.4–11)	< 0.001 (w)
troponin I, ng/L					
(NR, 0–53.5)					
ICU cohort		*n*=75	*n*=55	*n*=20	
Lymphocyte count, × 10⁹/L	72	0.85 (0.77–0.97)	0.77 (0.66–0.85)	0.98 (0.87–1.15)	0.009 (w)
(NR, 1.1–4.5)					
ALT, U/L	53	38 (32–51)	35 (31–52)	47 (29–73)	0.358 (w)
(NR, 0–35)					
Albumin, g/dL	28	4.1 (4.0–4.3)	4.1 (4–4.4)	4.3 (4–4.4)	0.494 (w)
(NR, 2.9–5.2)					
Creatine kinase, U/L	33	122 (102–176	122 (99–176)	135 (107–329)	0.679 (w)
(NR, 40–280)					
Creatinine, mg/dL	64	0.89 (0.8–1)	0.92 (0.82–1.16)	0.85 (0.73–0.97)	0.094 (w)
(NR, 0.7–1.3)					
D-dimer, ng/mL	26	1075 (752–1256)	1088 (745–1824)	758 (752–1647)	0.527 (w)
(NR, 0–500)					
Serum ferritin, ng/L	3	817 (520–852)	835 (817–852)	520	1.000 (w)
(NR, 22–322)					
Platelet count, ×10⁹/L	65	185 (170–224)	185 (165–225)	205 (175–265)	0.654 (w)
(NR, 150–370)					
IL-6, pg/mL	3	30.9 (11.5–244.0)	30.9 (11.5–244)	-	-
(NR, 0–3.4)					
Lactate dehydrogenase, U/L (NR, 100–190)	33	365 (323–451)	388 (332–520)	328 (240–401)	0.190 (w)
Procalcitonin, ng/mL	17	0.2 (0.1–0.5)	0.38 (0.12–0.93)	0.08 (0.06–0.16)	0.019 (w)
(NR, 0–0.5)					
C-reactive protein, mg/L	58	139.8 (109.8–188.4)	151 (118.1–203)	123.8 (25.1–169.7)	0.048 (w)
(NR, 0–5)					
Prothrombin time, s	63	11.2 (11.0–11.4)	11.3 (11–11.5)	11.1 (10.5–11.4)	0.177 (w)
High-sensitivity cardiac	7	24.6 (11.2–92.7)	15.4 (11.2–73.4)	80.2 (67.2–92.7)	0.190 (w)
troponin I, ng/L					
(NR, 0–53.5)					

Abbreviations: ALT, alanine aminotransferase; ICU, intensive care unit; IL, interleukin; IQR, interquartile range; NR, normal range.

**Table 5 jcm-09-01733-t005:** Risk factors associated with in-hospital death.

Variable	Multivariate OR (95% CI)	*p*-Value
Age, years	1.105 (1.087–1.124)	< 0.001
Sex, female	0.547 (0.375–0.794)	0.002
SatO_2_ < 90%	3.805 (2.539–5.723)	< 0.001
Lymphocyte count, ×10⁹/L	0.573 (0.356–0.879)	0.017
C-reactive Protein, mg/L	1.006 (1.003–1.008)	< 0.001

Abbreviations: SatO_2_, oxygen saturation.

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
