# Peer review of "A Cohort of Patients with COVID-19 in a Major Teaching Hospital in Europe"

_jcm, 2020, doi:10.3390/jcm9061733_

Round 1

Reviewer 1 Report

The authors describe a large Spanish cohort of patients with Covid-19 admitted to hospital. Factors associated with death is discussed. It adds to the current knowledge and confirm other reports and may be of interest to the readers.

Data are consequently presented in tables for all patients as well as for the subgroups of survivors vs. non-survivors. P-values should be presented in the tables in order to highlight statistically significant differences between survivors and non-survivors.

Mulivariate analysis should be presented when appropriate, e.g regarding symtoms at presentation, comorbidities as well as laboratory findings in relation to death.

General symptoms such as diarrhea, myalgia and headache seems to be more common among survivors, this is an interesting finding which should be discussed.

What was the cause of death for non-survivors? ARDS was reported in a minority of cases. What was the cause of death in the rest of the population of non-survivors?

Not all patients received treatment. What was the indication for treatment?

Was any kind of treatment associated with outcome or severe side effects? Was any arrythmias reported in patients treated with hydroxychloroquine or azithromycin?

Author Response

We were pleased to know that you are potentially interested in our manuscript. We would like to take this opportunity to express our sincere thanks to the reviewers for the queries raised and identification of areas of our manuscript that needed improvement. 

Comment: Data are consequently presented in tables for all patients as well as for the subgroups of survivors vs. non-survivors. P-values should be presented in the tables in order to highlight statistically significant differences between survivors and non-survivors.

Answer: We have added P-values according to reviewer suggestion in all the tables.

Comment: Mulivariate analysis should be presented when appropriate, e.g regarding symtoms at presentation, comorbidities as well as laboratory findings in relation to death.

Answer: Reviewer is right, but we believe that we already provide important descriptive clinical, laboratory and therapeutic data, that is the objective of our manuscript. We are waiting to finalize enrollment in our cohort to perform a more robust multivariant analysis. However, following reviewer suggestion, we present a preliminary multivariate analysis. We have included the information in the Method section (2.3) and the Results section (3), including a new table (Table 5).

Comment: General symptoms such as diarrhea, myalgia and headache seems to be more common among survivors, this is an interesting finding which should be discussed.

Answer: Thank you for the comment. We have included a paragraph in the Discussion section (4) highlighting this finding: “General symptoms as diarrhea, myalgia, headache or anosmia were more common among survivors, while respiratory symptoms such as dyspnea or sputum production were more prevalent among non-survivors, who have also a lower median of oxygen saturation on admission (90% vs 96%). In any case, it is necessary to confirm these findings in larger cohorts or metanalyses.”

Comment: What was the cause of death for non-survivors? ARDS was reported in a minority of cases. What was the cause of death in the rest of the population of non-survivors?

Answer: Thanks for the comment. Unfortunately, at this point we do not have the immediate cause of death in all of our patients. Although we are still analyzing medical records to try to adjudicate causes of death, we are afraid that in a substantial number of patients identifying with precision the immediate cause of death is going to be difficult.

Comment: Not all patients received treatment. What was the indication for treatment?

Answer: The indication for treatment has changed during the course of the epidemic. It has been based on the indications of the Spanish Ministry of Health, and the availability of treatments at all times. In addition to this, our hospital has participated in several clinical trials for which we cannot provide results in this manuscript (remdesivir, tocilizumab, sarilumab). We here included the use of treatment with potential antiviral effect, not included in a clinical trial and available in our hospital (hydroxychloroquine, azithromycin or lopinavir/ritonavir). It is also remarkable that at the beginning of the epidemic in our country, only symptomatic treatment was recommended.

Comment: Was any kind of treatment associated with outcome or severe side effects? Was any arrythmias reported in patients treated with hydroxychloroquine or azithromycin?

Answer: This is a good question. An analysis of this type (efficacy/safety of treatments used) is out of scope of this communication. The reason is that it is at very high risk of bias (specially selection bias) and could have relevant consequences on clinical practice. Therefore, we feel that a more in deep analysis is needed in order to control for all the bias that can be introduced.

Only for the reviewer's knowledge, we present here raw information on this subject in the subcohort of older patients (>65 years), that is a more homogeneous one in terms of median age and comorbidities: 

Reviewer 2 Report

The manuscript “A cohort of patients with COVID-19 in a major teaching hospital in Europe” by Borobia et.al., reports the systematic analysis of a cohort of 2236 COVID 19 patients hospitalized in a Madrid hospital. The data presented in the current study reconfirms that old age and co morbidities lead to poor prognosis of the disease. The cohort consists of a reasonable number of patients and presents a systematic documentation of the data.

There a few concerns and these concerns need to be addressed. The concerns are

Improvement of overall English is necessary. Lot of grammatical errors are there.

Reference should be within bracket.

Line 70: please mention where in Spain the first patient was reported, please consider reconstruction of the sentence, not making much sense.

Line72: with a maximum of 3419 new cases on March 30 -- what do you exactly mean by maximum new cases?

Line 76: How are you measuring the mortality rate here? What is the exact mortality rate? It cannot be 132, 140 or 190. And if these are the no of deaths then what are the no of infected person? There is no mention of it. Authors only wrote ‘per 1,00,000 inhabitants.

Line 120: what do you mean by pre-COVID bed?

Line 119: ‘occupancy with by COVID patients’ –write either with or by

Line 140: change ‘In the moment of the analysis’

Some comments on the treatment regimen (as in why a particular combination drug is chosen) should have been included.

Author Response

We were pleased to know that you are potentially interested in our manuscript. We would like to take this opportunity to express our sincere thanks to the reviewers for the queries raised and identification of areas of our manuscript that needed improvement. 

Comment: Improvement of overall English is necessary. Lot of grammatical errors are there.

Answer: Thank you for your comment. The manuscript has been now reviewed by an English style corrector.

Comment: Reference should be within bracket.

Answer:We have modified all references in the text, including brackets.

Comment: Line 70: please mention where in Spain the first patient was reported, please consider reconstruction of the sentence, not making much sense.

Answer: We have reconstructed the sentence and included the first patient reported location (line 69 and 70): The first infection in Spain was confirmed on January 31st, 2020, in the island of La Gomera [1].

Comment: Line72: with a maximum of 3419 new cases on March 30 -- what do you exactly mean by maximum new cases?

Answer: We have modified this sentence in order to clarify (line 72, 73 and 74): “The number of confirmed cases in Madrid was 58,819 as of April 25 (26.3% of cases in Spain) [3], reaching a peak of 3419 new cases on March 30

Comment: Line 76: How are you measuring the mortality rate here? What is the exact mortality rate? It cannot be 132, 140 or 190. And if these are the no of deaths then what are the no of infected person? There is no mention of it. Authors only wrote ‘per 1,00,000 inhabitants.

Answer: The reviewer is right, the term “rate” is incorrect. We have modified this sentence (line 76): “The healthcare systems of these areas are under very high stress and the cumulative Covid-19 mortality per 100,000 inhabitants since the start of the pandemic is high: 132 in Lombardia, 140 in New York City and 190 in Madrid (as of 25th of April).

We consider these figures much more robust than nº of infected persons. We consider that the cumulative number of infected persons (usually confirmed by PCR or rapid test) is much less reliable due to the differences in how they are detected in each country and region, as they depend on availability of test and case reporting policies putted in place. In any case this figures are publicly available in the web pages of the different countries or in aggregators as the carried out by John Hopkins University (https://coronavirus.jhu.edu/map.html).

Comment: Line 120: what do you mean by pre-COVID bed?

Answer: Pre-COVID is the period previous to the transformation of our hospital due to the epidemic. We have changed this sentence to clarify it: “Figure 1 shows occupancy by COVID-19 patients over time, with a peak of 1033 beds of which 106 were in ICU (compared to 30 ICU beds before COVID19 epidemic).”

Comment: Line 119: ‘occupancy with by COVID patients’ –write either with or by

Answer: We have changed it: “Figure 1 shows occupancy by COVID-19 patients over time…

Comment: Line 140: change ‘In the moment of the analysis’

Answer: We have changed it by: “At the time of the analysis

Comment: Some comments on the treatment regimen (as in why a particular combination drug is chosen) should have been included.

Answer:  Thank you for your comment. We have including a brief description in the discussion section (4): “Drug treatment has changed during the course of the epidemic. It has been based on the indications of the Spanish Ministry of Health, and the availability of treatments at all times. In addition to this, our hospital has participated in several clinical trials for which we cannot provide results in this manuscript (remdesivir, tocilizumab, sarilumab). We here have included the use of treatment with potential antiviral effect, not included in a clinical trial and available in our hospital (hydroxychloroquine, azithromycin or lopinavir/ritonavir). It is also remarkable that at the beginning of the epidemic in our country, only symptomatic treatment was recommended

Round 2

Reviewer 2 Report

The manuscript titled 'A Cohort of Patients with COVID-19 in a Major 2
Teaching Hospital in Europe' by Borobi AM et al is an improved version compared to the first submission. However, the reviewer suggests a careful checking of English language once again.

Author Response

Comment: The manuscript titled 'A Cohort of Patients with COVID-19 in a Major
Teaching Hospital in Europe' by Borobi AM et al is an improved version compared to the first submission. However, the reviewer suggests a careful checking of English language once again.

Answers: According to the reviewer suggestion, the manuscript has been reviewed by an english style corrector (Morote Traducciones - https://morote.net/en/home/ that has careful checked grammar and English style.